# Omni-Chat: Enhancing Spoken Dialogue Systems with Scalable Synthetic Data for Diverse Scenarios

## Abstract

With the rapid development of large language models, researchers have created increasingly advanced spoken dialogue systems that can naturally converse with humans. However, these systems still struggle to handle the full complexity of real-world conversations, including audio events, musical contexts, and emotional expressions, mainly because current dialogue datasets are constrained in both scale and scenario diversity. In this paper, we propose leveraging synthetic data to enhance the dialogue models across diverse scenarios. We introduce **ShareChatX**, the first comprehensive, large-scale dataset for spoken dialogue that spans diverse scenarios. Based on this dataset, we introduce **OmniChat**, a multi-turn dialogue system with a heterogeneous feature fusion module, designed to optimize feature selection in different dialogue contexts. In addition, we explored critical aspects of training dialogue systems using synthetic data. Through comprehensive experimentation, we determined the ideal balance between synthetic and real data, achieving state-of-the-art results on the real-world dialogue dataset DailyTalk. We also highlight the crucial importance of synthetic data in tackling diverse, complex dialogue scenarios, especially those involving audio and music. For more details, please visit our demo page at `https://sharechatx.github.io/`.

## 1 Introduction

With the rapid advancement of artificial intelligence, spoken dialogue systems (Jokinen & McTear, 2009; Ji et al., 2024) have emerged as a crucial branch of human-computer interaction. Many voice assistants, such as Siri (Hoy, 2018) and Cortana (Hachman, 2019), leverage automatic speech recognition (Yu & Deng, 2016) to transcribe speech into text and generate corresponding responses, enabling conversational capabilities. Driven by the progress in large language models (LLMs)(Touvron et al., 2023), modern spoken dialogue systems(OpenAI, 2024b) now possess enhanced reasoning and understanding abilities, allowing for more complex dialogue functions based on speech content. However, unlike traditional text-based dialogue systems Qin et al. (2023), spoken dialogue systems must also account for a wealth of multi-modal information beyond words. Significant efforts have been made to enhance multi-modal large language models for understanding various types of audio. Audio-Flamingo (Kong et al., 2024) has developed a text conversation dataset centered on audio events and music, enabling text-based dialogues built around these elements. Qwen-Audio 1/2 (Chu et al., 2023; 2024), trained on 520,000 hours of audio-related tasks, has equipped its models to comprehend speech, audio, music, and other full-scene audio inputs. EMOVA (Chen et al., 2024) introduces a framework that integrates spoken dialogue with multimodal tasks, enabling a spoken dialogue model that can "see, hear, and speak". Although these models have demonstrated some ability in handling spoken dialogues, the limitations in the scale and diversity of current dialogue datasets have led to the lack of a spoken dialogue system that can effectively understand speech emotions, audio events, or interpret background music in complex spoken dialogue scenarios.

Compared to the vast amounts of text-based conversational data available online (Sordoni et al., 2015), collecting spoken dialogue corpora presents significantly more challenges: **(1) Limited Scale of Spoken Dialogue Data.** Acquiring spoken dialogue data is both more complex and costly than gathering text data (Cieri et al., 2004), resulting in much smaller datasets. High-quality spoken data (especially data with multi-turn interactions and emotional complexity across different sce-

Table 1: Comparison of Spoken Dialogue Datasets. **E** indicates whether the dataset emphasizes emotional information, **A** indicates the presence of audio events in the dialogue, and **M** indicates the involvement of music. The dialogue data is derived from three scenarios: controlled environments (**Env**), in-the-wild collection (**Wild**), and AI generation (**AI-Gen**). **#Avg.** represents the average number of turns per dialogue. [†]All responses in E-chat200 are in text format, duration only includes speech on the query side. The dialogues in AF-Dialogue are all text-based, with duration reflecting only audio and music segments.

| Datasets | Scens. | | | Source | # Turns | #Dialog. | #Avg. | #Dur. |
|---|---|---|---|---|---|---|---|---|
| | E | A | M | | | | | |
| *Speech-to-Speech Dialogue Dataset* | | | | | | | | |
| IEMOCAP (Busso et al., 2008) | ✔ | ✗ | ✗ | Env | 10,039 | 151 | 66.48 | 12 |
| SwitchBoard (Godfrey et al., 1992) | ✗ | ✗ | ✗ | Wild | - | 2,500 | - | 250 |
| Fisher (Cieri et al., 2004) | ✗ | ✗ | ✗ | Wild | - | 11,699 | - | 1,960 |
| DSTC2 (Henderson et al., 2014) | ✗ | ✗ | ✗ | Wild | 23,354 | 1,612 | 14.49 | 32 |
| MELD (Poria et al., 2018) | ✔ | ✗ | ✗ | Wild | 13,000 | 1,433 | 9.07 | 14 |
| Expresso (Nguyen et al., 2023) | ✔ | ✗ | ✗ | Env | 2,400 | 391 | 6.14 | 47 |
| DailyTalk (Lee et al., 2023) | ✔ | ✗ | ✗ | Env | 23,774 | 2,514 | 9.46 | 22 |
| SpokenWOZ (Si et al., 2024) | ✗ | ✗ | ✗ | Env | 203,074 | 5,700 | 35.63 | 249 |
| StyleTalk (Lin et al., 2024) | ✔ | ✗ | ✗ | AI-Gen | 12,056 | 2,967 | 4.06 | 12 |
| **ShareChatX (ours)** | | | | | | | | |
| −*ShareChat-Emotion* | ✔ | ✗ | ✗ | AI-Gen | 588,174 | 80,152 | 7.34 | 672 |
| −*ShareChat-Audio* | ✔ | ✔ | ✗ | AI-Gen | 199,034 | 27,005 | 7.37 | 217 |
| −*ShareChat-Music* | ✔ | ✗ | ✔ | AI-Gen | 160,028 | 21,443 | 7.46 | 242 |
| −Overall | ✔ | ✔ | ✔ | AI-Gen | 947,236 | 128,600 | 7.37 | 1,130 |
| *Non-Speech-to-Speech Dialogue Dataset* | | | | | | | | |
| E-chat200 (Xue et al., 2023) | ✔ | ✗ | ✗ | AI-Gen | 356,000 | 178,000 | 2.00 | 193[†] |
| AF-Dialogue (Kong et al., 2024) | ✗ | ✔ | ✔ | AI-Gen | 657,600 | 82,200 | 8.00 | 228[†] |

narios (Lin et al., 2024)) is even more difficult to obtain. **(2) Lack of Copyright-Free Data.** Spoken dialogues inherently contain personal and biometric information, such as timbre, making anonymization difficult without degrading data quality. This raises privacy concerns when collecting and employing large scale spoken dialogue datasets. **(3) Lack of Scenario-Specific Spoken Dialogue Corpora.** Gathering spoken dialogue data from specific scenarios like emergencies or high-stakes environments is particularly challenging (Ao et al., 2024). These conversations often involve strong emotional reactions and unique audio conditions that are difficult to replicate or simulate. The lack of data from these specialized contexts limits the performance of dialogue systems.

In response to these challenges, we propose leveraging large-scale synthetic data to simulate complex dialogue scenarios, thus improving spoken dialogue models across diverse scenarios. Drawing on the powerful reasoning capabilities of large language model (OpenAI, 2024a), we generate dialogue scripts tailored to each scenario. These scripts are then converted into spoken dialogues using high-fidelity, controllable text-to-speech (TTS) model (Du et al., 2024). As shown in Table 1, we present **ShareChatX**, the first large-scale, comprehensive spoken dialogue dataset covering a broad range of scenarios, including -*Emotion* (involving complex emotional changes), -*Audio* (incorporating audio events), and -*Music* (featuring background music). We also introduce **OmniChat**, the first multi-turn spoken dialogue system designed to handle a wide range of scenarios. OmniChat features a heterogeneous feature fusion module called `Mix-Former`, engineered to optimize feature selection across different dialogue contexts. Furthermore, we conducted extensive experiments and analyses on various training methodologies to maximize the effectiveness of synthetic data in training spoken dialogue systems. This enabled us to determine the optimal balance between synthetic and real data, leading to state-of-the-art performance on the real-world spoken dialogue dataset DailyTalk (Lee et al., 2023). Our experiments also highlight the crucial importance of synthetic data in tackling complex dialogue scenarios, especially those involving audio and music. We will release the data and code at `https://sharechatx.github.io/`. Our main contributions are:

• We propose ShareChatX, the first large-scale, comprehensive spoken dialogue dataset covering a wide range of scenarios, including -*emotion*, -*audio*, and -*music*.

- We introduce OmniChat, the first multi-turn spoken dialogue system for diverse scenarios, with a heterogeneous feature fusion module to optimize expert feature selection across varied scenarios.
- We discussed various details involved in training spoken dialogue models with synthetic data, and explored best practices for building effective spoken dialogue systems based on synthetic data.
- We achieve state-of-the-art performance on the real-world spoken dialogue dataset DailyTalk and other complex dialogue scenarios, demonstrating the importance of scaleable synthetic data.

## 2 RELATED WORKS

### 2.1 SPOKEN DIALOGUE DATASETS

Intelligent dialogue has long been a central focus of artificial intelligence research. To advance the development of dialogue systems, researchers have collected vast amounts of open-domain text dialogue data from public platforms like Twitter (Sordoni et al., 2015) and Weibo (Shang et al., 2015) for training purposes. However, the scale of spoken dialogue datasets has remained limited, hindering the progress of spoken dialogue systems. Early efforts (Godfrey et al., 1992; Cieri et al., 2004) involved constructing datasets by recruiting participants to record spoken dialogues, but this approach was resource-intensive and costly. Later, researchers (Poria et al., 2018) turned to publicly available resources such as TV shows to collect spoken dialogue data, resulting in the creation of numerous real-world spoken dialogue datasets that were systematically annotated with emotional information. With the maturation of large language models (LLMs) (OpenAI, 2024a), researchers have begun synthesizing spoken dialogue data using AI-generated methods. For instance, Kong et al. (2024) adopted ChatGPT-4 (OpenAI, 2024b) to generate AF-Dialogue, a textual dialogue dataset set within specific audio event or music scenarios. Lin et al. (2024) employed large language model (OpenAI, 2024a) and the controllable TTS models to create StyleTalk, a dataset focused on capturing different emotions to generate contextually appropriate responses.

However, obtaining large-scale, scenario-specific spoken conversation data remains challenging, limiting the scale and diversity of existing spoken dialogue datasets. To address this, we introduce ShareChatX, the first large-scale, omni-scenario synthetic spoken conversation dataset. ShareChatX covers a broad range of conversations, including those focused on speech emotion (*-Emotion*), audio events (*-Audio*), and music understanding (*-Music*).

### 2.2 LARGE AUDIO-LANGUAGE MODEL

With the development of large language models, increasingly powerful audio language models have emerged, leveraging extensive training corpora to achieve comprehensive audio understanding capabilities. SpeechGPT (Zhang et al., 2023) integrates discrete speech units into the LLM, making it the first speech-centric large language model. Qwen-Audio1/2 (Chu et al., 2023; 2024) build the first comprehensive large-scale audio model on more than 30 audio-related tasks, including speech recognition, speech translation, and audio event detection. Salmonn (Tang et al., 2023) addresses the problem of task overfitting in audio models by introducing more complex story generation tasks. Building on the understanding of audio, a series of spoken dialogue models have been developed to promote more intelligent human-computer interaction. Audio-Flamingo (Kong et al., 2024) creates an audio event-centric text dialogue dataset, enabling multi-turn, audio-focused text conversations. StyleTalk (Lin et al., 2024) focuses on emotional conversation tasks and introduces the first spoken dialogue model that responds with different emotional tones.

However, due to limitations in training data, most current spoken dialogue models are restricted to question answering purpose (Chu et al., 2024) or experiments on small-scale datasets (Lin et al., 2024). To address this gap, we propose using synthetic data to enhance the performance of spoken dialogue models across various scenarios and introduce OmniChat, the first multi-turn spoken dialogue model designed for diverse contexts.

## 3 SHARECHATX

As shown in Figure 1, the ShareChatX dataset is divided into three sub-datasets: *-Emotion*, *-Audio*, and *-Music*, each characterized by distinct metadata. *-Emotion* includes dialogue samples with rich

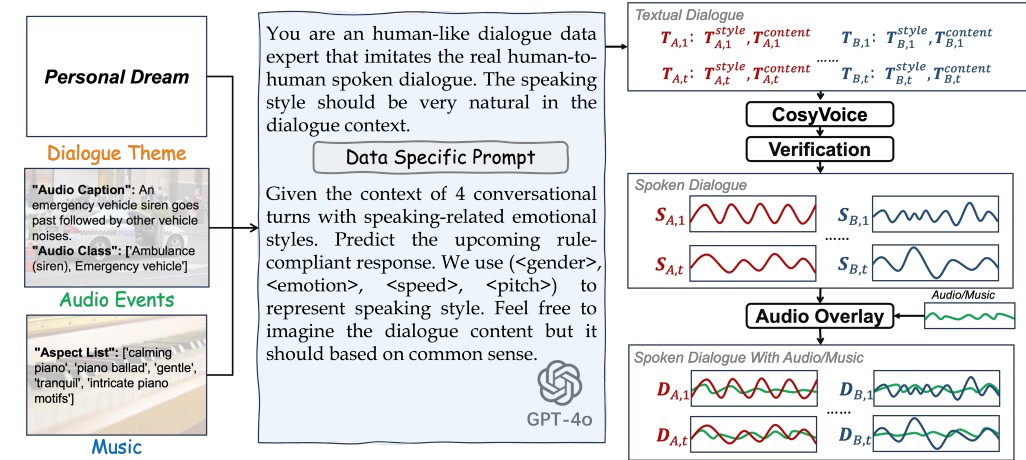

Figure 1: **Overview for Crafting our ShareChatX Dataset.** First, text dialogue scripts $T_i = \{T_i^{style}, T_i^{content}\}$ are generated using large language models, with data-specific prompts tailored for the three subsets: *-emotion*, *-audio*, and *-music*. Next, spoken dialogue data $S_i$ is synthesized using controllable text-to-speech synthesis model (CosyVoice-Instruct), incorporating style parameters such as gender, pitch, speed, and emotion. To ensure the quality of the generated data, both model-based and manual verification processes are applied. Finally, audio events and music are integrated into the dialogues, with specific methods for handling temporary and continuous sounds.

emotional expression, *-Audio* focuses on conversations centered around audio events, and *-Music* features samples incorporating background music. Below, we provide a detailed explanation of the dataset annotation process:

**Textual Dialogue Scripts.** Leveraging the powerful reasoning capabilities of large language models (OpenAI, 2024a), we create textual dialogue scripts tailored to different topics and scenarios using detailed prompt templates. In this process, we instruct the model to first generate $N$ rounds of historical dialogue, followed by responses and emotions that match the contextual flow. The dialogue topics for *-emotion* subset are generated with large language models, the audio descriptions for *-audio* subset are derived from AudioCaps (Kim et al., 2019), and the music information for *-music* subset is sourced from MusicCaps (Agostinelli et al.). For further details, see Appendix E.2.

**Spoken Dialogue.** In the textual script generation step, we generated the style parameters $T_i^{style}$ (gender, pitch, speed, emotion) and the corresponding text content $T_i^{content}$ for each sentence $T_i$. Using these style parameters and the text content, we employed the open-source controllable TTS model, CosyVoice-Instruct (Du et al., 2024), to synthesize the corresponding speech $S_i$.

**Dialogue Verification.** To ensure the quality of the voice conversation data, we implemented a dual verification method combining model-based and manual checks. Since each voice clip in the conversation is generated separately, we used a speaker diarization model (Plaquet & Bredin, 2023) to confirm that the same speaker's voice maintained consistent timbre. Additionally, we applied an ASR model (Radford et al., 2023) to ensure that the word error rate (WER) across all samples did not exceed 5%. For each conversation, we attempted synthesis up to 10 times until the entire conversation met the required standards. Finally, manual inspection was conducted to verify that each sample adhered to the logic of natural human conversation.

**Audio/Music Integration.** For *ShareChat-Audio* and *ShareChat-Music*, we overlay the corresponding audio and music onto the spoken dialogue data. For *-audio* subset, a large language model (LLM) is used to determine whether the event is temporary or continuous. Temporary audio events, such as a door slamming or a phone ringing, are short-lived sounds that occur briefly and are spliced before the first voice segment. In contrast, continuous audio events, like background chatter or street noise, are prolonged sounds that persist over time and are looped as background sound throughout the conversation. For the *-music* subset, we randomly apply two different methods to combine the music with the dialogues. To ensure the authenticity of the final dialogue, all audio and music components are overlaid according to Petermann et al. (2022) when combined with speech. For further details, see Appendix E.1.

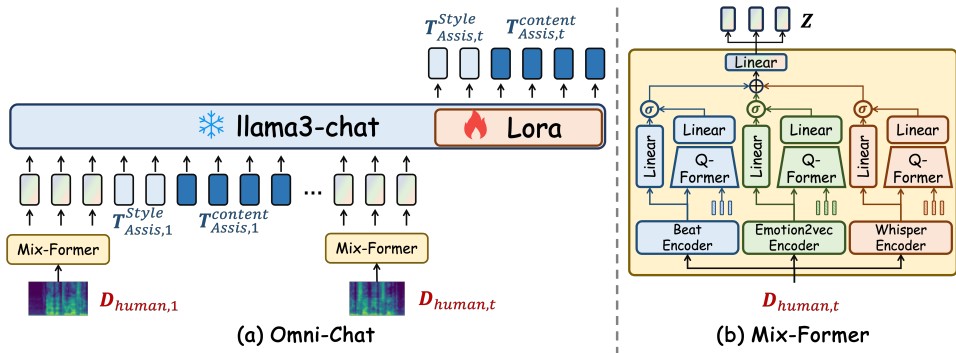

(a) Omni-Chat          (b) Mix-Former

Figure 2: **Overview of OmniChat**. (a) OmniChat predicts the $t$-th response $\mathbf{T}_{Assis,t}$ by using the previous $t$ dialogues $\mathbf{D}_{human,1}, \cdots, \mathbf{D}_{human,t}$ and $t-1$ responses $\mathbf{T}_{Assist,1}, \cdots, \mathbf{T}_{Assist,t\text{-}1}$ as context. OmniChat concurrently predicts both the Style $\mathbf{T}_{Assist,t}^{style}$ and Content $\mathbf{T}_{Assist,t}^{content}$ of the response. (b) `Mix-Former` leverages Q-Former to independently represent different expert features, thereby enhancing the ability to capture the nuances of each aspect of the dialogue segment.

## 4 SPOKEN DIALOGUE SYSTEM

The spoken dialogue system aims to generate an appropriate response $\mathbf{D}_{Assist,T}$ based on the contextual information from the spoken dialogue sequence $\{\mathbf{D}_{human,1}, \mathbf{D}_{human,2}, \ldots, \mathbf{D}_{human,T}\}$ and the preceding response sequence $\{\mathbf{D}_{Assist,1}, \mathbf{D}_{Assist,2}, \ldots, \mathbf{D}_{Assist,T-1}\}$, where $T$ represents the total number of dialogue turns. Following previous work (Lin et al., 2024), each response is represented by two components: $\mathbf{T}_{Assist,i}^{style}$, which conveys the emotional tone, and $\mathbf{T}_{Assist,i}^{content}$, which represents the speech content. These components can then be fed into controllable TTS models (Du et al., 2024) to synthesize highly expressive and contextually appropriate responses $\mathbf{D}_{Assist,i}$.

### 4.1 OMNICHAT

As illustrated in Figure 2, subfigure (a) depicts our proposed OmniChat, a multi-turn spoken dialogue model built upon a large language model. This model is capable of generating the most appropriate response acoustic style $\mathbf{T}_{Assist,t}^{style}$ and content $\mathbf{T}_{Assist,t}^{content}$ for various voice inputs and dialogue scenarios. In subfigure (b), after extracting features using multiple expert models, the heterogeneous fusion module MIX-FORMER is employed to produce the final voice feature input. The detailed introduction is as follows:

**Multi-Expert Audio Feature Extraction** In spoken dialogue, capturing acoustic features beyond just the speech content is crucial. To model these features, we employ multiple expert models, each specializing in a different dimension of the speech. For the speech content, we utilize Whisper model (Radford et al., 2023) to extract speech content features $\mathbf{F}_s$, trained with weak supervision on large-scale speech corpora, represented as $\mathbf{F}_i^s = \{\mathbf{F}_{i,1}^s, \cdots, \mathbf{F}_{i,n}^s\} = $ `Whisper-Encoder`$(\mathbf{D}_{human,i})$. For emotional information, we use Emotion2vec (Ma et al., 2023), a speech emotion representation model trained with self-supervision on extensive emotional speech datasets, which captures the emotional nuances of the speech $\mathbf{F}_i^e = \{\mathbf{F}_{i,1}^e, \cdots, \mathbf{F}_{i,n}^e\} = $ `emotion2vec`$(\mathbf{D}_{human,i})$. To enable the model to understand broader non-speech elements, such as audio events and music, we incorporate the Beat model (Chen et al., 2023) as a non-speech audio feature extractor $\mathbf{F}_i^b = \{\mathbf{F}_{i,1}^b, \cdots, \mathbf{F}_{i,n}^b\} = $ `Beat-Encoder`$(\mathbf{D}_{human,i})$. Since the feature frame rates of these audio expert encoders are consistent, the three expert features can be temporally aligned as $\{(\mathbf{F}_{i,j}^s, \mathbf{F}_{i,j}^e, \mathbf{F}_{i,j}^b) \mid j \in [1, N]\}$, where $N$ is the number of frames in each audio feature.

**Mix-Former for Heterogeneous Fusion**

The importance of different features can vary significantly across dialogue system scenarios. For example, beat features are essential in music-related environments but may interfere with emotion-centric dialogues. To address this, we propose a heterogeneous feature fusion module called MIX-FORMER, as shown in Figure 2 (b), which integrates diverse expert features while minimizing interference.

For each expert feature, we use an attribute-specific window-level Q-Former to align audio and language between frozen audio encoders and a frozen large language model (LLM). The expert features $\mathbf{F}_i^s \in \mathbb{R}^{N \times D_s}, \mathbf{F}_i^e \in \mathbb{R}^{N \times D_e}, \mathbf{F}_i^b \in \mathbb{R}^{N \times D_b}$, corresponding to the audio segment, are segmented into windows of size $L$. The Q-Former at the window level uses a fixed number of $K$ trainable queries $\mathbf{Q^s}, \mathbf{Q^e}, \mathbf{Q^b}$ to encode the features stacked in each window into $K$ hidden features:

$$\mathbf{H}_i^s = \mathtt{Q\text{-}Former}(\mathbf{Q^s}, \mathbf{F}_i^s), \quad \mathbf{H}_i^e = \mathtt{Q\text{-}Former}(\mathbf{Q^e}, \mathbf{F}_i^e), \quad \mathbf{H}_i^b = \mathtt{Q\text{-}Former}(\mathbf{Q^b}, \mathbf{F}_i^b), \quad (1)$$

where $\mathbf{H}_i^s \in \mathbb{R}^{[N \times K/L] \times D_s}, \mathbf{H}_i^e \in \mathbb{R}^{[N \times K/L] \times D_e}, \mathbf{H}_i^b \in \mathbb{R}^{[N \times K/L] \times D_b}$ represent window-level attribute features. To adapt to different scenarios, we introduce a weight module that assigns weights $\mathbf{w}$ to each feature using three linear layers:

$$\mathbf{w}_{i,l}^s = \sigma(\mathtt{Linear}(\mathbf{H}_{i,l}^s)), \quad \mathbf{w}_{i,l}^e = \sigma(\mathtt{Linear}(\mathbf{H}_{i,l}^e)), \quad \mathbf{w}_{i,l}^b = \sigma(\mathtt{Linear}(\mathbf{H}_{i,l}^b)), \quad (2)$$

where $l$ is the $l$-th window-level feature and $\sigma(\cdot)$ is the sigmoid function. The weighted expert features are concatenated as: $\mathbf{H}_i = \mathtt{concat}(\mathbf{w}_i^s \mathbf{H}_i^s, \mathbf{w}_i^e \mathbf{H}_i^e, \mathbf{w}_i^b \mathbf{H}_i^b)$, where $\mathtt{concat}(\cdot)$ is the frame-by-frame concatenation operation along the feature dimension, $\mathbf{H}_i \in \mathbb{R}^{[N \times K/L] \times [D_s + D_e + D_b]}$. This concatenated feature is then linearly projected to align with the input space $\mathbf{Z}_i$.

## 4.2 TRAINING METHOD

During training, we freeze all parameters of the audio feature extractor and LLM, focusing solely on training the Q-Former and the LoRA adapters, which adjust the query and value weight matrices in the self-attention layers of the LLM. The entire model is optimized using the multi-turn dialogue loss, which is calculated as follows:

$$L_{\text{dialogue}} = -\sum_{t=1}^{T} \sum_{j=1}^{m} \log p(\mathbf{T}_t^j | \mathbf{Z}_{1:t}, \mathbf{T}_{1:t-1}, \mathbf{T}_t^{1:j-1}), \quad (3)$$

where $T$ is the total number of dialogue turns, $m$ is the number of tokens in the $t$-th turn's response, $\mathbf{T}_t^j$ is the $j$-th token in the response for the $t$-th turn, $\mathbf{Z}_{1:t}$ represents the audio features up to the $t$-th turn, and $\mathbf{T}_{1:t-1}$ refers to the tokens from all previous turns, while $\mathbf{T}_t^{1:j-1}$ denotes the preceding tokens within the same turn. This loss function ensures the model learns to generate contextually appropriate responses over multiple dialogue turns, leveraging both the dialogue history and the audio features.

## 5 EXPERIMENTS

## 5.1 IMPLEMENTATION DETAILS

We adopt the Llama-3.1-8B-Instruct model (Dubey et al., 2024) as the backbone LLM. All audio data are resampled to 16 kHz for consistency. In the windowed Q-Former, we set $K = 1$, resulting in a single trainable query, and use $L = 17$, which corresponds to approximately 0.33 seconds per window. The models are trained for 30,000 steps with a batch size of 48 on eight A800 GPUs. For more detailed training information, refer to Appendix D. To evaluate model performance, we conducted experiments on two datasets: DailyTalk (Lee et al., 2023) and our proposed ShareChatX. For testing, we randomly selected a test set from each subset of DailyTalk and ShareChatX, ensuring that the training and test sets were non-overlapping. Following previous studies (Lin et al., 2024), we employed both quantitative and qualitative metrics to evaluate model performance. The quantitative evaluation was divided into two aspects: content and style. For content evaluation, we utilized widely recognized text generation metrics, including vocabulary-level scores such as BLEU (Papineni et al., 2002), ROUGE-L (Lin, 2004), and METEOR (Banerjee & Lavie, 2005), as well as semantic-level metrics like BERTScore (Zhang et al., 2019). For style evaluation, we computed weighted F1 scores for speaking emotion. In addition to the quantitative metrics, we conducted qualitative analyses using GPT-based metric (Yang et al., 2024) and manual evaluation. The detailed prompt template for GPT evaluation can be found in Appendix D.3. For a dialogue with $T$ turns, we use the previous $T - 1$ turns as context and predict only the response for the $T$-th turn.

Table 2: Performance Comparison of Various Spoken Dialogue Systems on the DailyTalk Dataset. The content metrics include **@B** (BLEU), **@R** (ROUGE-L), **@M** (METEOR), and **@BS** (BERTScore). The Style metrics include **@F1**$_e$ for emotion prediction accuracy.

| Methods | @B | @R | @M | @BS | F1$_e$ | GPT-eval | MOS |
|---|---|---|---|---|---|---|---|
| *ASR-Based Spoken Dialogue System* | | | | | | | |
| StyleTalk (Lin et al., 2024) | 2.01 | 9.42 | 10.95 | 82.82 | 49.63 | 3.51 | 3.42±0.23 |
| FunAudioLLM (SpeechTeam, 2024) | 2.65 | 12.53 | 11.82 | 84.76 | 61.02 | 3.82 | 3.85±0.18 |
| *Direct Spoken Dialogue System* | | | | | | | |
| Audio-Flamingo (Kong et al., 2024) | 1.47 | 5.01 | 10.23 | 83.94 | - | 2.35 | 2.53±0.25 |
| SpeechGPT (Zhang et al., 2023) | 1.42 | 7.85 | 9.42 | 84.11 | - | 2.68 | 2.45±0.32 |
| Qwen-Audio (Chu et al., 2023) | 2.04 | 7.43 | 11.21 | 84.33 | - | 3.01 | 3.23±0.18 |
| Salmonn (Tang et al., 2023) | 2.32 | 11.78 | 11.56 | 85.47 | - | 3.41 | 3.05±0.22 |
| Qwen2-Audio (Chu et al., 2024) | 3.03 | 12.81 | 13.89 | 86.14 | - | 4.01 | 3.87±0.25 |
| OmniChat (ours) | 3.54 | 12.63 | 12.57 | 86.24 | 71.87 | 3.96 | 3.97±0.22 |
| OmniChat + Real Data (ours) | **4.95** | **12.95** | **14.24** | **86.99** | **75.46** | **4.15** | **3.99±0.18** |

Table 3: Performance comparison of various methods for spoken dialogue systems on the ShareChatX datasets. The content metrics include **@B** (BLEU), **@R** (ROUGE-L), **@M** (METEOR), and **@BS** (BERTScore). The Style metrics include **@F1**$_e$ for emotion prediction accuracy.

| Methods | ShareChat-Emotion | | | | | ShareChat-Audio | | | | | ShareChat-Music | | | | |
|---|---|---|---|---|---|---|---|---|---|---|---|---|---|---|---|
| | @B | @R | @M | @BS | @F1$_e$ | @B | @R | @M | @BS | @F1$_e$ | @B | @R | @M | @BS | @F1$_e$ |
| *ASR-Based Spoken Dialogue System* | | | | | | | | | | | | | | | |
| FunAudioLLM | 3.2 | 14.9 | 18.8 | 86.9 | 46.7 | 3.3 | 12.0 | 12.9 | 86.0 | 41.9 | 3.0 | 12.0 | 12.4 | 86.2 | 49.2 |
| *Direct Spoken Dialogue System* | | | | | | | | | | | | | | | |
| Qwen-Audio | 3.0 | 8.2 | 12.2 | 84.3 | - | 3.0 | 7.3 | 11.9 | 84.0 | - | 2.9 | 9.0 | 11.7 | 84.1 | - |
| Salmonn | 2.9 | 11.8 | 11.4 | 86.1 | - | 3.6 | 10.1 | 11.2 | 85.6 | - | 2.9 | 10.5 | 11.1 | 86.1 | - |
| Qwen2-Audio | 3.1 | 14.2 | 17.4 | 86.7 | - | 3.6 | 12.2 | 13.2 | 87.2 | - | 3.0 | 12.2 | 13.4 | 87.2 | - |
| OmniChat (ours) | **6.2** | **20.0** | **18.9** | **88.1** | **57.2** | **6.0** | **18.7** | **17.4** | **87.3** | **51.5** | **4.7** | **17.7** | **15.8** | **87.8** | **69.1** |

## 5.2 MAIN RESULTS

**Comparison on Real-World Spoken Dialogue.** As shown in Table 2, we evaluated the performance of spoken dialogue models on the DailyTalk real-world spoken dialogue dataset. The models were categorized into ASR-Based Spoken Dialogue Systems, which rely on ASR-transcribed text, and Direct Spoken Dialogue Systems, which generate responses directly from speech input. **(1) Response Content:** OmniChat demonstrated superior performance across all content-related metrics, particularly when fine-tuned with real data. For instance, OmniChat + Real Data achieved the highest METEOR score of 14.24 and a BERTScore of 86.99, outperforming direct models like Qwen2-Audio (METEOR: 13.89, BERTScore: 86.14). These results highlight the importance of synthetic data for responding in real-world dialogue scenarios. **(2) Emotion Prediction Accuracy:** OmniChat also significantly outperforms all other models in terms of emotion prediction, with OmniChat + Real Data achieving an $F1_e$ score of 75.46, far exceeding the best ASR-based model, FunAudioLLM (61.02). Even without fine-tuning, OmniChat achieved an impressive 71.87, demonstrating its superior ability to detect and generate emotionally appropriate responses. Since real-world data may lack diverse emotional interactions, synthetic data helps bridge this gap by enriching the dialogue corpus with dynamic emotional shifts, which further supports model training.

**Comparison on Diverse Complex Dialogue Scenes.** As shown in Table 3, the analysis of the ShareChatX dataset (-*Emotion*, -*Audio*, -*Music*) demonstrates the significant improvements OmniChat offers in dialogue generation and emotion prediction for complex scenarios. OmniChat consistently excels in content generation and accurately predicts emotional shifts, highlighting its effectiveness in handling multi-modal dialogues. It is worth noting that while Qwen2-Audio improved its BLEU (from 3.1 of -*emotion* to 3.6 of -*Audio*), key metrics like ROUGE-L and METEOR dropped significantly, indicating that recognizing audio events alone is insufficient for generating coherent dialogue in complex scenarios. OmniChat, by leveraging large-scale multi-modal synthetic dialogue data, maintains strong performance even in challenging environments. Its ability to integrate multi-modal information enhances both dialogue generation and emotion recognition, emphasizing the importance of comprehensive data for improving system performance.

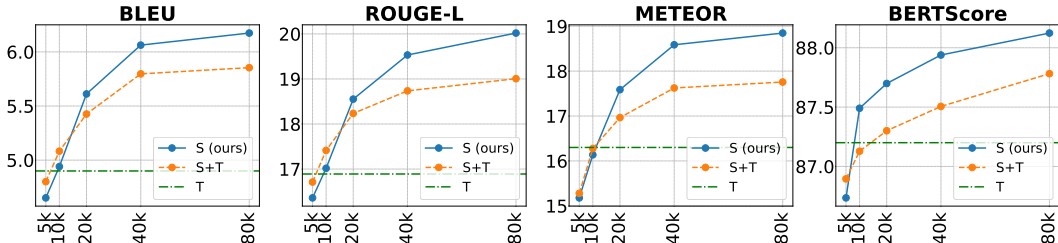

Figure 3: **Performance comparison of dialogue systems trained with varying data scales on the ShareChatX-Emotion.** **T** denotes text input, **S+T** denotes both speech and ASR-transcription input, and **S (ours)** represents our method utilizing only speech as input. The numbers on the horizontal axis represent the scale of the dialogue data used during training.

## 5.3 HOW DOES DATA SCALE IMPACT THE SPOKEN DIALOGUE MODELS?

Spoken dialogue models based on large language models must learn the mapping between speech and text from scratch, and the scale of training data plays a crucial role in their performance. However, *what scale of data is sufficient to support the training of effective spoken dialogue models?* To explore this question, we conducted a comparative analysis of the three most commonly used input modalities for dialogue models, as shown in Figure 3: (1) text-based dialogue models (using text as input, represented by the green line), (2) ASR-based spoken dialogue models (utilizing ASR transcriptions along with speech input, represented by the orange line), and (3) direct spoken dialogue models (relying solely on speech input, represented by the blue line). The following analysis highlights the key findings as the dataset scale ranges from 5K to 80K samples.

**Speech Models Surpass Text Models (5K-10K)**    At the 5K-10K data scale, models incorporating speech input (either with or without ASR transcriptions) begin to outperform text-based models. For example, the BLEU score of the direct speech model improves from 4.65 at 5K to 4.94 at 10K, while the text-based model lags behind with a BLEU score of 4.86. Speech data, which contains not only semantic content but also emotional cues, allows the model to capture richer information than text alone, leading to better performance as the dataset size increases.

**Direct Speech Model Outperforms ASR-based Model (10K-20K)**    Between the 10K and 20K data scale, the direct speech model (without ASR text) begins to outperform the ASR-based model. For instance, the METEOR score of the direct speech model reaches 17.59 at 20K, while the ASR-based model trails slightly behind at 16.96. At this scale, the ASR transcriptions no longer provide additional useful information; in other words, this amount of data is sufficient for the model to learn the mapping from speech to semantics from scratch.

**Textual Input Becomes Redundant (20K-80K)**    As the dataset size increases further (20K-80K), the performance of the speech-only model continues to improve, while the performance of the ASR-based model plateaus. For example, in the BLEU metric, the direct speech model improves from 6.06 at 20K to 6.17 at 80K, whereas the ASR-based model shows diminishing returns, rising only from 5.79 to 5.85. This suggests that as the model is trained on larger datasets, speech alone is sufficient to capture all necessary information, including emotional cues and context. In contrast, the text input becomes redundant, as it lacks the multimodal information present in speech, such as tone, intonation, and emotion. This redundancy not only fails to improve performance but can also hinder the model by introducing unnecessary complexity. For instance, at the 80K data scale, the ROUGE-L score of the speech-only model reaches 20.02, while the model using both speech and ASR text achieves only 19.01.

## 5.4 OPTIMAL SAMPLING RATIOS OF SYNTHETIC AND REAL DATA IN SPOKEN DIALOGUE MODELS

Researchers have demonstrated that achieving optimal performance across various tasks requires balancing synthetic and real data during training. Synthetic data enhances model robustness, while real data ensures alignment with the target domain's distribution. Yet, the question arises: *what is the ideal sampling ratio for spoken dialogue models?* (Please note that the importance of data scale has been demonstrated in subsection 5.3. The experiments in this subsection focus solely on the sampling rate, with the synthetic data scale fixed at 80K.)

Table 4: Performance comparison of models trained with varying mixup ratios of synthetic and real data on the DailyTalk dataset. $\alpha$ represents the frequency of synthetic data used during training.

| Ratio | BLEU | ROUGE-L | METEOR | BERT Score | $F1_e$ |
|---|---|---|---|---|---|
| $\alpha$=0.0 | 3.54 | 12.63 | 12.57 | 86.24 | 71.87 |
| $\alpha$=0.1 | **5.07** | **13.29** | 14.17 | 85.70 | 71.05 |
| $\alpha$=0.2 | 4.95 | 12.95 | **14.24** | **86.99** | **75.46** |
| $\alpha$=0.3 | 3.94 | 12.04 | 13.84 | 86.42 | 74.71 |
| $\alpha$=0.4 | 3.88 | 11.97 | 13.25 | 85.90 | 73.55 |
| $\alpha$=0.5 | 3.73 | 11.63 | 13.05 | 85.26 | 72.36 |
| $\alpha$=1.0 | 3.90 | 12.32 | 13.22 | 86.02 | 70.66 |

To explore this, we experimented with various sampling ratios, as shown in Table 4, to determine the optimal balance between synthetic and real data: **(1) Low Ratio ($\alpha = 0.1$) Ensures Lexical Consistency.** At a sampling ratio of $\alpha = 0.1$ (one synthetic sample for every ten training samples), the model achieved a BLEU of 5.07 and a ROUGE-L of 13.29, outperforming models trained exclusively on real data ($\alpha = 1.0$) or synthetic data ($\alpha = 0.0$). This indicates that incorporating a small proportion of synthetic data helps the model achieve better consistency at the word level, while real data ensures alignment with natural spoken dialogues. **(2) Moderate Ratio ($\alpha = 0.2$) Achieves Sentence-Level Consistency.** Further increasing the proportion of synthetic data improved the model's ability to generate semantically coherent responses. At a sampling ratio of $\alpha = 0.2$, the model's $F1_e$ score increased by 4.41 compared to $\alpha = 0.1$, demonstrating that this ratio allows the model to achieve optimal performance at the sentence level in terms of meaning and emotion control. **(3) Excessive Ratio ($\alpha > 0.2$) Leads to Performance Decline.** When the ratio of synthetic data exceeded $\alpha = 0.2$, performance in real conversation scenarios began to decline. For instance, the ROUGE-L dropped by 0.91 when $\alpha$ increased from 0.2 to 0.3, indicating that an excessive reliance on synthetic data can hinder the model's ability to generalize to real-world conversations. Based on these findings, a sampling ratio of $\alpha = 0.2$ provides the ideal balance, achieving optimal performance in real-world dialogue scenarios.

## 5.5 MULTI-EXPERT SPEECH FEATURE FOR SPOKEN DIALOGUE SYSTEMS.

As shown in Table 5, we present a performance comparison of different expert feature selection strategies on the ShareChat-Music dataset, focusing on the role of Mix-Former (M-F) and three expert features: speech features ($F_s$), emotion features ($F_e$), and beat features ($F_b$).

The experiment shows that simply adding expert features without proper integration can lead to performance degradation. For example, when speech ($F_s$) and emotion ($F_e$) features were combined without Mix-Former, the METEOR score dropped to 15.4, compared to 15.8 when only speech features ($F_s$) were used.

Table 5: Performance Comparison of Different Expert Feature Selection Strategies on *ShareChat-Music*. M-F stands for Mix-Former.

| $F_s$ | $F_e$ | $F_b$ | **M-F** | @B | @R | @M | @BS | @F1$_e$ |
|---|---|---|---|---|---|---|---|---|
| ✔ | | | | 4.65 | **17.8** | **15.8** | 87.5 | 66.7 |
| ✔ | ✔ | | | **4.68** | 17.6 | 15.4 | 87.5 | 68.8 |
| ✔ | ✔ | ✔ | | 4.63 | 17.7 | 15.6 | 86.3 | 69.0 |
| ✔ | ✔ | ✔ | ✔ | **4.68** | 17.7 | **15.8** | 87.8 | 69.1 |

(Methods | SHARECHAT-MUSIC)

However, when Mix-Former was applied, the model successfully combined multiple expert features, leading to improved results. With speech, emotion, and beat features ($F_s$, $F_e$, $F_b$) processed through Mix-Former, the model achieved the highest METEOR score of 15.8 and the best BERTScore of 87.8, demonstrating its ability to effectively capture and integrate diverse expert feature.

## 6 CONCLUSION

Spoken dialogue systems have been hindered by the scarcity of large-scale, high-quality spoken dialogue data. To address this challenge, we introduced the use of synthetic datasets to enhance the performance of dialogue models. In this paper, we presented ShareChatX, the first large-scale dataset covering diverse, complex scenarios such as emotional dialogues, audio events, and music. Through extensive experimentation, we determined the optimal balance between real and synthetic data, as well as the required data size for training spoken dialogue models.

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

## A USE OF LLM

We use llm to generate data and evaluate models.

## B LIMITATIONS

The spoken dialogue system we proposed, Omnichat, currently focuses on generating the most appropriate reply content and emotions but still relies on a controllable TTS model to synthesize speech for replies. However, the research in this article emphasizes the understanding capabilities of spoken dialogue systems, and the conclusions drawn can also serve as a reference for end-to-end spoken dialogue models that directly generate speech. In the future, we will explore the application of synthetic data in developing end-to-end spoken dialogue systems.

## C  ETHICAL DISCUSSION

Spoken dialogue systems developed using public data may face risks such as inappropriate guidance or offensive language. Due to the complexity and diversity of conversations in public datasets, it can be challenging to determine whether the content poses risks, such as encouraging criminal behavior. In contrast, dialogue systems developed using synthetic data can better ensure ethical consistency in conversation content. Additionally, this paper is intended solely for academic research and does not result in commercial products, so the ethical risks are minimal at present. We plan to explore how to further reduce the risk of accidental guidance in voice dialogue systems in the future.

## D  MORE EXPERIMENTAL DETAILS

### D.1  DETAILS FOR DIALOGUE ON DAILYTALK

In mixed training with real and synthetic data, we sample from both datasets at a specific sampling rate $\alpha$. For each training instance, a random number $\mu$ is drawn between 0 and 1. If $\mu < \alpha$, the model selects samples from the synthetic data for training. If $\mu \geq \alpha$, the samples are selected from the real data for training. We randomly selected 220 samples from DailyTalk as the test set. We will open-source the test set partitions in this work to facilitate comparison in future studies.

### D.2  DETAILS FOR DIALOGUE ON COMPLEX SCENARIOS

For *ShareChat-Emotion*, we train the model directly on the *ShareChat-Emotion* dataset and proceed to evaluate it. For *ShareChat-Audio* and *ShareChat-Music*, we leverage a model pre-trained on *ShareChat-Emotion* and fine-tune it on these two subsets to better adapt the model for specific complex scenarios. Both the pre-training on *ShareChat-Emotion* and the fine-tuning on the two subsets are conducted for 30,000 steps each. We have 3,731 dialogues for the *-emotion* test set, 1,555 for the *-audio* test set, and 1,243 for the *-music* test set.

### D.3  PROMPT TEMPLATE FOR GPT-EVAL

As illustrated in Figure 4, we present the template utilized for GPT-based evaluation (GPT-eval).

## E  MORE DETAILS ABOUT SHARECHATX.

### E.1  TEMPORARY AND CONTINUOUS AUDIO EVENTS

We use GPT-4 to determine whether audio events are temporary or continuous, which guides how we concatenate audio and spoken dialogues. Specifically, the prompt template for this step is shown in Figure 6.

### E.2  PROMPT TEMPLATE

**ShareChat-Emotion**  For *ShareChat-Emotion*, we utilized a large language model (LLM) to randomly generate 521 dialogue topics. Below are several examples of these topics to provide a clearer understanding of the dialogue content: *Artistic hobbies*, *Regrets from the past*, *Dealing with difficult people*, *Communication styles*, and *The culture of food*. In Figure 5, we present the emotion distribution for *ShareChat-Emotion*. The detailed prompt template for *ShareChat-Emotion* is shown in Figure 10.

**ShareChat-Audio**  For ShareChat-Audio, we used the captions of audio events in AudioCaps (Kim et al., 2019) as keywords to generate conversations. To prevent interference with the dialogues, we employed PANNs (Kong et al., 2020) to identify and remove all human voice events. The detailed prompt template for *ShareChat-Audio* is shown in Figure 11.

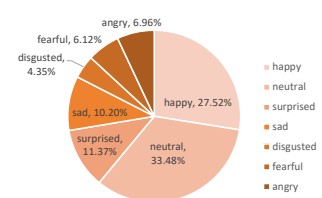

Figure 5: Emotion distribution in *ShareChat-Emotion*.

You are an evaluator assessing the quality of responses generated by a spoken dialogue system. Each response will be scored on a scale of 1 to 5 based on the following criteria:

1: The response is contextually irrelevant and expresses inappropriate emotion.
2: The response is partially relevant to the context but feels unnatural, with inappropriate emotion.
3: The response is partially relevant to the context and expresses appropriate emotion.
4: The response is contextually relevant and natural, mentioning the attribute, and expresses appropriate emotion.
5: The response is contextually relevant, smooth, natural, and expresses emotions that perfectly align with the scenario.

Here is the context and the generated response. Please evaluate the response based on the above criteria and provide a score along with a brief explanation for your rating.

**Example Input**

**Context:**
"User: I just got promoted at work, but I feel so overwhelmed lately."

**Generated Response:**
"Congratulations on your promotion! It's normal to feel overwhelmed with new responsibilities. Take things step by step, and you'll get the hang of it."

**Expected Output**

**Score**: 5
**Explanation**: The response is contextually relevant, natural, and emotionally supportive, perfectly aligning with the user's scenario and emotions.

Figure 4: The Prompt Template for GPT-eval.

> I will provide you with a description of the background sound and ask you to determine what may have happened based on the background sound and analyze whether this will happen again in a short period of time?
>
> The background sound is described as **[The wind is blowing, insects are singing, and rustling occurs].**
>
> All you have to do is reply "This will happen again" or "This will not happen again".

Figure 6: The prompt template to determine whether audio events are temporary or continuous.

**ShareChat-Music** For ShareChat-Music, we used the aspect list from the audio clips in MusicCaps (Kim et al., 2019) as keywords to generate dialogues. The aspect list includes detailed information such as music type, instrument type, emotion, and other characteristics of each piece of music. The detailed prompt template for *ShareChat-Music* is shown in Figure 12.

# F  MORE EXPERIMENTAL RESULTS

## F.1  DIALOGUE RESULTS IN REAL-WORLD

We present cases of the model output on the Dailytalk dataset in Figure 7, showcasing Omnichat's logical reasoning and expression capabilities in real-world scenarios. Compared to FunAudioLLM and Qwen2-Audio, our model generates more natural and reasonable responses, with accurate emotional expressions tailored to the conversation.

## F.2  DIALOGUE RESULTS IN COMPLEX SCENARIOS

Cases in Figure 8 and Figure 9 present the results of Omnichat compared to other baselines on Sharechat-Emotion, Sharechat-Audio, and Sharechat-Music datasets. On the Sharechat-Emotion dataset, we evaluated the model's diverse outputs when presented with identical text but with varying emotional undertones in the dialogue corpus. The enthusiastic replies in response to a positive attitude and the comforting words when faced with a melancholic tone, demonstrating model's adeptness at discerning and responding to emotional subtleties

The results in Figure 9 demonstrate that the model not only effectively captures bird chirping sounds in the background environment but also understands the music in the background and is capable of expressing its own perspective. Compared to the baseline, the model's superior understanding of background sounds indicates that the extensive synthetic data and the novel heterogeneous feature fusion module have endowed it with more versatile conversational capabilities.

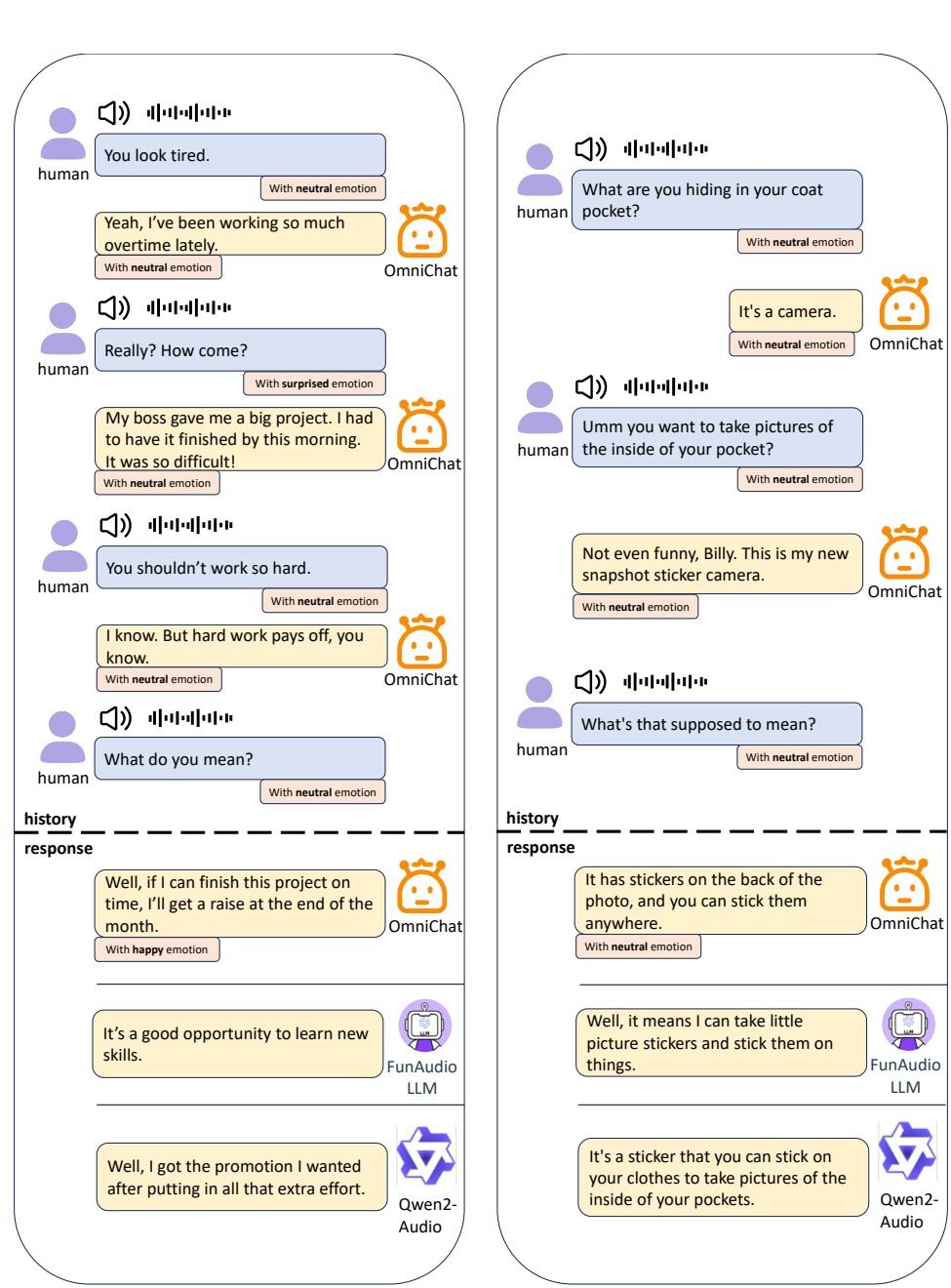

Figure 7: Dialogue results samples on the DailyTalk Dataset.

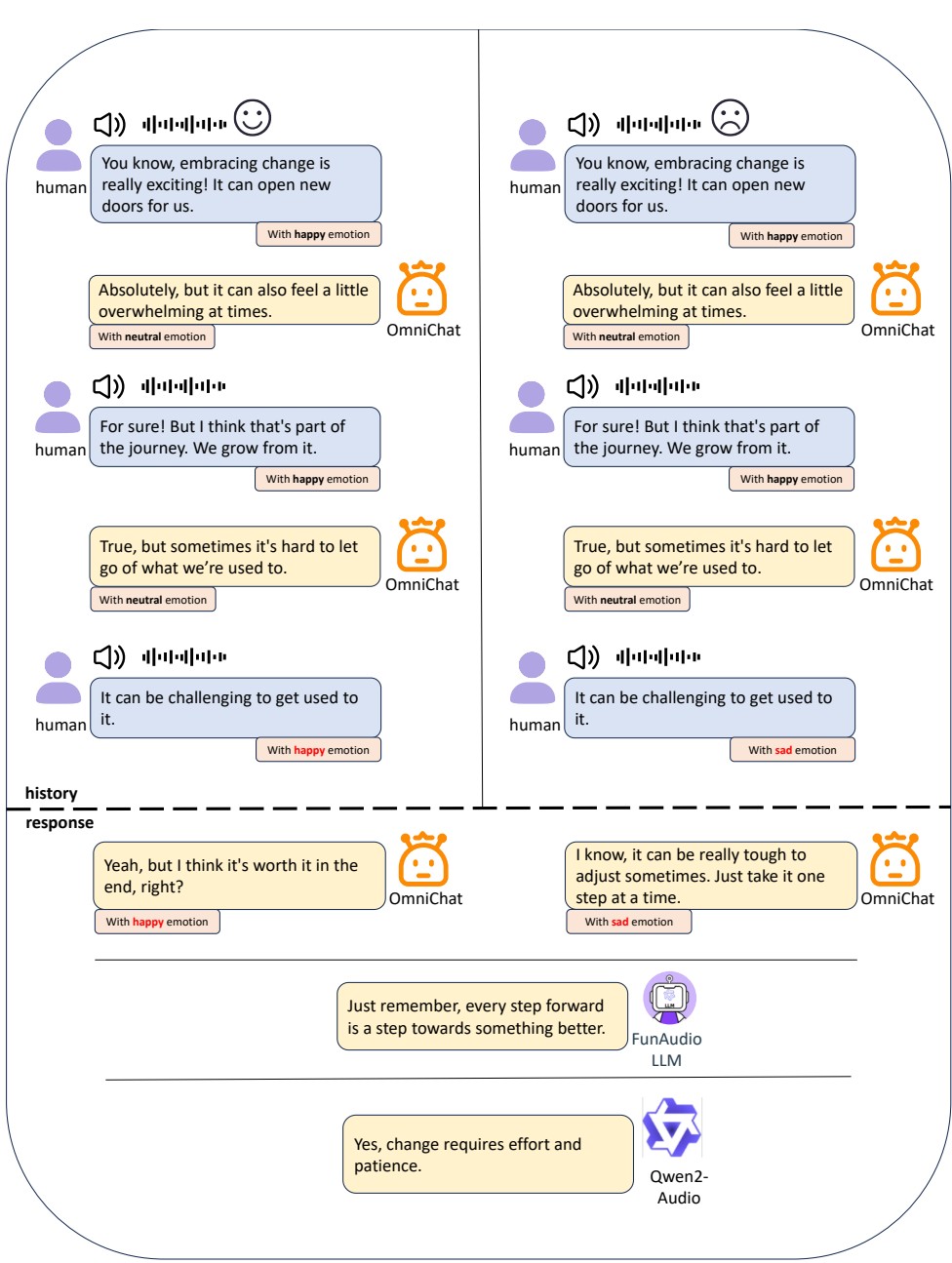

Figure 8: Dialogue results samples on the ShareChat-Emotion Dataset.

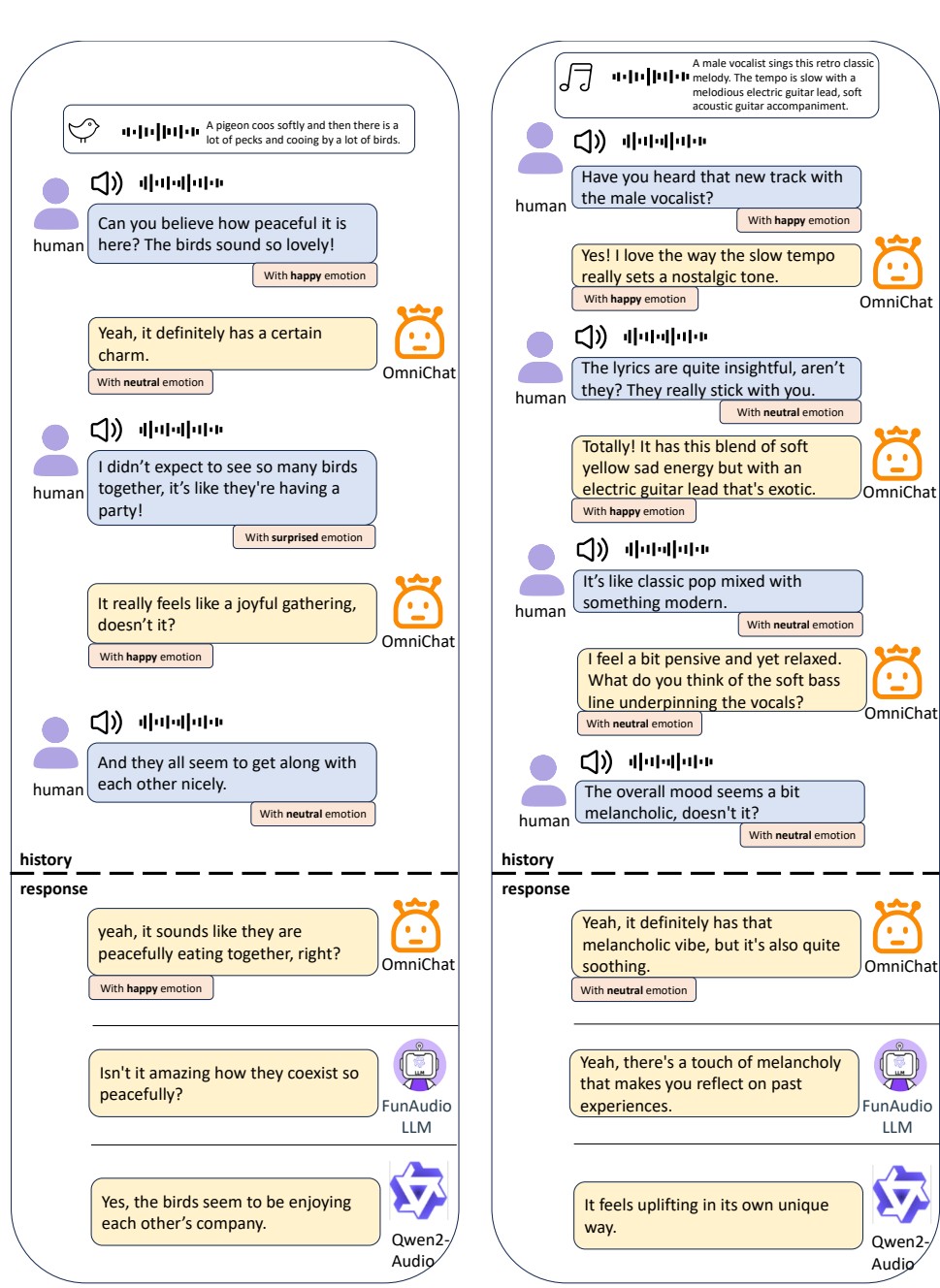

Figure 9: Dialogue results samples on the ShareChat-Audio and ShareChat-Music Dataset.

**Prompt For *ShareChat-Emotion***

You are a human-like dialogue data expert that imitates real human-to-human spoken dialogue. The speaking style should be very natural in the dialogue context.

**Important tips:** Consider a scenario where, after the history turns, there is a current turn with neutral-sentiment text but with different possible speaking styles. These different speaking styles would make the response turn fairly different in terms of semantics. Just one sentence for each turn. The sentence should be spoken and spontaneous, not too formal.

**Please strictly follow these rules:**
1. We use special tokens < > to represent the class type that you have to generate. Do not include < > in the output.
2. You can only use these styles to represent speaking style (<gender>, <emotion>, <speed>, <pitch>). Important: Do not use any class that is not defined below!
3. Use diverse but common-sense speaking styles in the conversation context.
4. The text of the current turn should be in neutral sentiment, and the response turn should carefully consider the current turn and respond naturally, not just copying the current turn style.
5. There are two speakers (A and B) in the dialogue. Speakers A and B should have a back-and-forth interaction.
6. Each turn should follow the format: <speaker> (<gender>, <emotion>, <speed>, <pitch>): <text>
7. The order of turns is history turns -> current turn -> upcoming response.
8. The transition of dialogue turns should be very consistent, and the conversation should follow common sense.
9. The dialogue should contain emotional variation.
10. The output valid dictionary format is as follows:
    {
    "history turns": [ "<speaker> (<gender>, <emotion>, <speed>, <pitch>): <text>", ...],
    "current_turn": "<speaker>: <text>",
    "current_turn_style_1": "(<gender>, <emotion>, <speed>, <pitch>)",
    "current_turn_style_2": "(<gender>, <emotion>, <speed>, <pitch>)",
    "current_turn_style_3": "(<gender>, <emotion>, <speed>, <pitch>)",
    "response_of_current_style_1": "<speaker> (<gender>, <emotion>, <speed>, <pitch>): <text>",
    "response_of_current_style_2": "<speaker> (<gender>, <emotion>, <speed>, <pitch>): <text>",
    "response_of_current_style_3": "<speaker> (<gender>, <emotion>, <speed>, <pitch>): <text>"
    }
11. Output a valid dictionary example so that it can be parsed as a dictionary.
12. For <speaker>, remember to use only A or B.
13. For <gender>, remember to use only "male" or "female."
14. For <emotion>, remember to use only "neutral," "happy," "angry," "sad," "surprised," "fearful," or "disgusted." Do not use other words for emotions!
15. For <speed>, remember to use only "slow," "normal," or "fast."
16. For <pitch>, remember to use only "low," "normal," or "high."

Given the context of **[4/6/8]** conversational turns with speaking-related emotional styles, there are current turns with the EXACT SAME WORDS in 3 different styles, respectively. Predict the upcoming rule-compliant response. We use (<gender>, <emotion>, <speed>, <pitch>) to represent speaking style. The dialogue topic is **[Favorite Book].** Feel free to imagine the dialogue content, but it should be based on common sense.

Figure 10: The prompt template for *ShareChat-Emotion*. The green words are alternative key words.

**Prompt For *ShareChat-Audio***

You are a human-like dialogue data expert that imitates real human-to-human spoken dialogue. The speaking style should be very natural in the dialogue context.

**Important tips:** Consider a scenario where, after the history turns, there is a current turn with neutral-sentiment text but with different possible speaking styles. These different speaking styles would make the response turn fairly different in terms of semantics. Just one sentence for each turn. The sentence should be spoken and spontaneous, not too formal.

**Please strictly follow these rules:**

1. We use special tokens < > to represent the class type that you have to generate. Do not include < > in the output.
2. You can only use these styles to represent speaking style (<gender>, <emotion>, <speed>, <pitch>). Important: Do not use any class that is not defined below!
3. Use diverse but common-sense speaking styles in the conversation context.
4. The text of the current turn should be in neutral sentiment, and the response turn should carefully consider the current turn and respond naturally, not just copying the current turn style.
5. There are two speakers (A and B) in the dialogue. Speakers A and B should have a back-and-forth interaction.
6. Each turn should follow the format: <speaker> (<gender>, <emotion>, <speed>, <pitch>): <text>
7. The order of turns is history turns -> current turn -> upcoming response.
8. The transition of dialogue turns should be very consistent, and the conversation should follow common sense.
9. The dialogue should contain emotional variation.
10. The output valid dictionary format is as follows:
    {
    "history turns": [ "<speaker> (<gender>, <emotion>, <speed>, <pitch>): <text>", ...],
    "current_turn": "<speaker>: <text>",
    "current_turn_style_1": "(<gender>, <emotion>, <speed>, <pitch>)",
    "current_turn_style_2": "(<gender>, <emotion>, <speed>, <pitch>)",
    "current_turn_style_3": "(<gender>, <emotion>, <speed>, <pitch>)",
    "response_of_current_style_1": "<speaker> (<gender>, <emotion>, <speed>, <pitch>): <text>",
    "response_of_current_style_2": "<speaker> (<gender>, <emotion>, <speed>, <pitch>): <text>",
    "response_of_current_style_3": "<speaker> (<gender>, <emotion>, <speed>, <pitch>): <text>"
    }
11. Output a valid dictionary example so that it can be parsed as a dictionary.
12. For <speaker>, remember to use only A or B.
13. For <gender>, remember to use only "male" or "female."
14. For <emotion>, remember to use only "neutral," "happy," "angry," "sad," "surprised," "fearful," or "disgusted." Do not use other words for emotions!
15. For <speed>, remember to use only "slow," "normal," or "fast."
16. For <pitch>, remember to use only "low," "normal," or "high."

Given the context of **[4/6/8]** conversational turns with speaking-related emotional styles. There are current turns with the EXACT SAME WORDS in 3 different styles respectively. make sure that style complies with rules 12 through 15. Predict the upcoming rule-compliant response. We use (<gender>, <emotion>, <speed>, <pitch>) to represent speaking style. The dialog occurs in the background sound of **[A child shouts, and an adult male speaks, while an emergency vehicle siren sounds with the horn blowing]**, feel free to imagine events and dialog that might occur in this background sound, but be consistent with common sense.

Figure 11: The prompt template for *ShareChat-Audio*. The green words are alternative key words.

**Prompt For *ShareChat-Music***

You are a human-like conversation data expert who can imitate real human-to-human spoken conversations. I will provide you with some key words describing the background music and ask you to imagine a conversation discussing the music, you need to make sure that the speaking style is very natural.

**Important tips:** Consider a scenario that after the history turns, there is a current turn with neutral-sentiment text but with different possible speaking styles, the different current speaking styles would make the response turn fairly different in terms of semantics. Just one sentence for each turn. The sentence is spoken and spontaneous not too formal.

**Please strictly follow these rules:**

1. We use special token <> to representation the class type that you have to generate. Do not have <> in the output.
2. You can only use these styles for representation speaking style (<gender>, <emotion>, <speed>, <pitch>). Important, do not use other class that is not defined below!!!
3. Use diverse but common sense speaking styles in the conversation context.
4. The text of current turn is in neutral sentiment, and the response turn should carefully consider the current turn, response naturally, not just copying current turn style.
5. There are two speakers (A and B) in the dialogue. The speaker A and B talk with back and forth interaction.
6. Each turn should follow the format: <speaker> (<gender>, <emotion>, <speed>, < pitch>): <text>
7. The order of turns is history turns -> current turn -> upcoming response.
8. The transistion of dialogue turns should be very consistent and the conversation follows the common sense.
9. The dialouge contains emotional variation.
10. The output valid dictionary format is as below:
    {
    "history turns": [ "<speaker> (<gender>, <emotion>, <speed>, <pitch>): <text>", ...],
    "current_turn": "<speaker>: <text>",
    "current_turn_style_1": "(<gender>, <emotion>, <speed>, <pitch>)",
    "current_turn_style_2": "(<gender>, <emotion>, <speed>, <pitch>)",
    "current_turn_style_3": "(<gender>, <emotion>, <speed>, <pitch>)",
    "response_of_current_style_1": "<speaker> (<gender>, <emotion>, <speed>, <pitch>): <text>",
    "response_of_current_style_2": "<speaker> (<gender>, <emotion>, <speed>, <pitch>): <text>",
    "response_of_current_style_3": "<speaker> (<gender>, <emotion>, <speed>, <pitch>): <text>",
    }
11. Output the valid dictionary example, so that it can be parse as dictionary.
12. For <speaker>, remember to use only A or B.
13. For <gender>, remember to use only "male" and "female".
14. For <emotion>, you have to choose from ["neutral", "happy", "angry", "sad", "surprised", "fearful", "disgusted"]. Don't use other words!!!!!!
15. For <speed>, remember to use only "slow", "normal" or "fast".
16. For <pitch>, remember to use only "low", "normal" or "high".

Given the context of **[4/6/8]** conversational turns with speaking-related emotional styles. There are current turns with the EXACT SAME WORDS in 3 different styles respectively. make sure that style complies with rules 12 through 15. Predict the upcoming rule-compliant response. We use (<gender>, <emotion>, <speed>, <pitch>) to represent speaking style. The background music can be described with the keywords **['amateur recording', 'finger snipping', 'male mid range voice singing', 'reverb']** and you are free to imagine any common-sense dialog about this music.

Figure 12: The prompt template for *ShareChat-Music*. The green words are alternative key words.

