# OpenReview forum: "OmniChat: Enhancing Spoken Dialogue Systems with Scalable Synthetic Data for Diverse Scenarios"
_ICLR.cc/2026/Conference — Submitted to ICLR 2026_

### Official Review · Reviewer_ApeQ · 2025-10-25

**Soundness:** 3
**Presentation:** 2
**Contribution:** 3
**Rating:** 6
**Confidence:** 4

**Summary:**

The paper introduces ShareChatX, a synthetic spoken dialogue dataset spanning three scenario families; Emotion, -Audio, and -Music and OmniChat, a multi‑turn dialogue model that fuses Whisper / Emotion2Vec / BEAT features via a window‑level Q‑Former (“Mix‑Former”) while freezing the audio encoders and the LLM (LoRA is used only on the LLM). The pipeline and the model are depicted in Fig. 1–2. The paper reports SOTA on DailyTalk, analyzes data scale (5K→80K) and synthetic:real sampling (optimal around α≈0.2), and ablates expert‑feature fusion on ShareChatX.

**Strengths:**

The work delivers a comprehensive synthetic spoken-dialogue pipeline across three scenario families with explicit quality controls (ASR WER≤5%, diarization), and a practical modeling recipe that freezes the LLM/audio encoders and learns a Mix-Former fusion over Whisper/Emotion2Vec/BEAT to jointly predict style and content per turn. Results show consistent gains on DailyTalk and all ShareChatX subsets, plus actionable analyses: scaling curves where speech-only surpasses ASR-based/text-only with sufficient data, and a useful synthetic:real mixing guideline around α≈0.2 (with synthetic fixed at 80K). These findings are directly relevant for practitioners building spoken dialogue systems under data scarcity.

**Weaknesses:**

- Originality
    - Novelty concentrates on the Mix-Former fusion; there’s no head-to-head with strong alternatives (e.g., gating/MoE/late-fusion cross-attn) under matched budgets.
    - The claimed “wide range of scenarios” seems to exclude overlap/backchannels, limiting realism.
- Quality
    - Manual evaluation (Sec. 5.1) is under-specified: rater N, recruitment, instructions, tasks, agreement/statistics (CI/effect sizes) are missing.
    - Per-row training-data specs are unclear for Tables 2–3 (sources/amounts/sampling including α, steps, seeds); details are scattered in Appx. D.1–D.2.
    - α reporting mismatch: Table 2 doesn’t say whether “OmniChat + Real Data” used α=0.2, nor the #seeds/variance.
    - Table 3 lacks a “+Real Data” row, so cross-domain generalization on synthetic distributions isn’t quantified.
    - Design sensitivity is absent: choices K=1 and L=17 (~0.33 s) are not ablated; music/ambient contexts may need longer windows.
- Clarity
    - In Contribution, “large-scale” should be quantified following Table 1 totals.
    - Modality pipelines (T vs. S+T vs. S) need a concise block diagram and loss definitions; Fig. 3 alone doesn’t specify input routing.
    - Provide a single data-spec table in §5.2 that aligns with Tables 2–3 (summarizing Appx. D.1–D.2).
    - Conclusion should briefly summarize OmniChat and bring limitations (Appx. B) into the main text.
- Significance
    - The empirical message that textual input becomes redundant at ≥20K is compelling but may be over-generalized (language/domain/OOD).
    - Practical uptake would be stronger with fusion-baseline comparisons and OOD tests (or at least a discussion of expected robustness).

**Questions:**

- Manual eval protocol: Can you report rater N, recruitment criteria, instructions, sample counts per rater, agreement measure, and statistical tests/CI used in §5.1?
- Training data per row (Tables 2–3): Will you include a unified table listing data types/amounts/sampling (α)/steps/seeds for each model/config in the main text?
- α in Table 2: Was “OmniChat + Real Data (ours)” trained with α=0.2? If not, which α, and what are the #seeds/variance? How does this align with §5.4’s conclusion?
- Fusion alternatives & design sensitivity: Can you add either (a) a brief K/L ablation, or (b) a compact comparison to one strong fusion baseline (e.g., cross-attn late fusion) under the same compute?
- Overlap/backchannels: Do any ShareChatX splits include overlapping speech/backchanneling? If not, can you state this explicitly and outline a plan or estimate for adding such cases?

---

### Official Review · Reviewer_TLvw · 2025-10-26

**Soundness:** 2
**Presentation:** 3
**Contribution:** 2
**Rating:** 2
**Confidence:** 4

**Summary:**

This paper introduces ShareChatX, large-scale dataset for spoken dialogue dataset covering a broad range of scenarios, including emotion, audio, and music. This paper also proposes OmniChat, the multi-turn spoken dialogue system for diverse scenarios, with a heterogeneous feature fusion module to optimize expert feature selection across varied scenarios.

**Strengths:**

-	This paper introduces a large-scale synthetic dataset for spoken dialogue including emotion, audio, and music.
-	This paper also introduces a spoken dialogue system, OmniChat, which showed better results on DailyTalk and ShareChatX than other several models.

**Weaknesses:**

-	The OmniChat model introduced in Table 3 outperformed baseline models, but this may be due to in-domain evaluation, as only this model is trained and evaluated on the ShareChatX dataset.
-	The model shown to have the highest performance in Table 2, OmniChat + RealData, is trained on not only ShareChatX but also a different split of the evaluation data, DailyTalk. If discussing the importance of including “real data” in addition to synthetic data, as in Table 4, it would be better to evaluate using a dataset different from DailyTalk, as the current evaluation is influenced by being in-domain.
-	Regarding  ShareChatX dataset, there are only evaluations conducted after training, and no analysis of the quality of the dataset itself. It would be better to analyze whether there are any issues with the synthetic dialogue data.

**Questions:**

-	The case where α=0.2 is listed as OmniChat+RealData in Table 2, but looking at Table 4, it seems that the most balanced value has been selected. Ideally, the value of α should also be determined using a different split.
-	Is the human evaluation mentioned in Section 5.1 referring to the MOS evaluation in Table 2? And also is it an evaluation of sound quality? It would be better to include human evaluation not only for reference-based evaluations like BLEU and LLM-based evaluations like GPT-eval.

---

### Official Review · Reviewer_6SJm · 2025-10-30

**Soundness:** 3
**Presentation:** 2
**Contribution:** 2
**Rating:** 2
**Confidence:** 5

**Summary:**

This paper introduces ShareChatX, a large-scale synthetic spoken dialogue dataset covering diverse scenarios (emotion, audio events, music), and OmniChat, a multi-turn spoken dialogue system with a heterogeneous feature fusion module (Mix-Former). The authors argue that synthetic data can address the limitations of existing spoken dialogue datasets in terms of scale and diversity. They present extensive experiments, including ablations on data scale and synthetic/real data ratios, and claim state-of-the-art results on the DailyTalk dataset.

**Strengths:**

1. The paper presents ShareChatX, a large and diverse synthetic spoken dialogue dataset, which could be a valuable resource for the community if released.
2. The authors conduct a wide range of experiments, including ablation studies on data scale, synthetic/real data ratios, and feature fusion strategies.

**Weaknesses:**

1. The proposed module and overall system architecture are incremental over existing multi-modal fusion approaches. The technical novelty is limited, and the paper does not provide sufficient analysis to justify the need for the new module.
2. The evaluation on real-world data is limited in scope and depth. The evaluation primarily uses the DailyTalk dataset, which is not a standard benchmark for AudioLLM models. For a fair and meaningful comparison with other AudioLLM approaches, it would be more appropriate to use widely adopted public datasets. This limits the credibility and generalizability of the reported results.
3. The paper’s main contribution is based on synthetic data, but it does not convincingly demonstrate that models trained on such data generalize well to real-world scenarios. The improvements on real datasets (e.g., DailyTalk) are marginal and may be due to overfitting to synthetic patterns.

**Questions:**

1. Can you provide more evidence (e.g., human evaluation, qualitative analysis) that models trained on ShareChatX generalize to real-world spoken dialogue, beyond the limited automatic metrics on DailyTalk?

2. How do you ensure that the synthetic dialogues (especially for complex scenarios like music and audio events) are realistic and representative of real conversations?

---

### Official Review · Reviewer_sGYK · 2025-11-12

**Soundness:** 2
**Presentation:** 3
**Contribution:** 3
**Rating:** 4
**Confidence:** 3

**Summary:**

The paper introduces ShareChatX, a large-scale synthetic spoken dialogue dataset spanning emotion, audio events, and music, and OmniChat, a multi-turn spoken dialogue model with a heterogeneous fusion module (Mix-Former) that achieves SOTA results by optimally leveraging synthetic and real data.

**Strengths:**

1. The paper proposes the first comprehensive synthetic spoken dialogue dataset that covers emotion, audio events and music, with 128K dialogues and nearly 1M turns and rich metadata, addressing data scale and diversity limitations.

2. The paper carefully studies the impact of data scale, optimal sampling ratios of synthetic and real data, and conducts ablation studies for the Mix-Former module.

**Weaknesses:**

1. The reported optimal sampling ratio $\alpha=0.2$ is derived under fixed synthetic scale and a single real dataset. Therefore, generality across tasks/domains is uncertain.

2. Mix-Former (M-F) yields only small gains over stacking/single features in Section 5.5, suggesting limited incremental benefit under the reported setup and metrics.

3. The paper’s evaluation of the "speech" modality is insufficient. It relies primarily on text-based content metrics (BLEU/ROUGE-L/METEOR/BERTScore) and a single emotion-weighted $F_1$, without dedicated speech-side understanding/quality evaluation.

**Questions:**

1. In Section 5.5 on ShareChat-Music, Mix-Former shows modest gains. Can you provide ablations on -Emotion and -Audio and showcase the necessity of using M-F?

2. Your study reports an optimal synthetic–real sampling ratio of $\alpha=0.2$ on DailyTalk. Could you provide cross-dataset robustness and sensitivity analyses to assess whether this ratio generalizes beyond DailyTalk?

---

### Meta-Review · Area_Chair_7dPm · 2025-12-15

**Summary:**

The paper’s idea of a large synthetic spoken dialogue corpus (ShareChatX) plus a lightweight multi turn model (OmniChat with Mix Former) is valuable and practically motivated. Reviewers agree the dataset and empirical analyses could help practitioners. However, concerns converge on limited technical novelty, insufficient evaluation on real world/standard benchmarks, in domain bias around DailyTalk, and under specified human/speech side evaluations. It is challenging to judge true generality and incremental technical value.

**Reviewer Concerns:**

No rebuttals from authors

**Reviewer Scores:**

No chance.

---

### Decision · Program_Chairs · 2026-01-26

Reject